# Impact of Nutritional Assessment on the Clinical Outcomes of Patients with Non-*albicans* Candidemia: A Multicenter Study

**DOI:** 10.3390/nu13093218

**Published:** 2021-09-16

**Authors:** Yi-Chien Lee, Yong-Chen Chen, Jann-Tay Wang, Fu-Der Wang, Min-Han Hsieh, Ing-Moi Hii, Yu-Lin Lee, Mao-Wang Ho, Chun-Eng Liu, Yen-Hsu Chen, Wei-Lun Liu

**Affiliations:** 1Department of Internal Medicine, Fu Jen Catholic University Hospital, Fu Jen Catholic University, New Taipei City 243, Taiwan; cimedin7@yahoo.com.tw; 2School of Medicine, College of Medicine, Fu Jen Catholic University, New Taipei City 242, Taiwan; yongchenchen0824@gmail.com; 3Master Program of Big Data in Biomedicine, College of Medicine, Fu Jen Catholic University, New Taipei City 242, Taiwan; 4Department of Internal Medicine, National Taiwan University Hospital, Taipei 100, Taiwan; 14bcr@yahoo.com.tw; 5Institute of Infectious Diseases and Vaccinology, National Health Research Institutes, Tsu-Nan County 350, Taiwan; 6Division of Infectious Diseases, Department of Internal Medicine, Taipei Veterans General Hospital, Taipei 112, Taiwan; fdwang@vghtpe.gov.tw; 7School of Medicine, National Yang-Ming University, Taipei 112, Taiwan; 8Division of Infectious Diseases, Department of Internal Medicine, Kaohsiung Medical University Hospital, Kaohsiung Medical University, Kaohsiung 807, Taiwan; 970388kmuh@gmail.com (M.-H.H.); d810070@kmu.edu.tw (Y.-H.C.); 9Division of Infectious Diseases, Department of Internal Medicine, Changhua Christian Hospital, Changhua 500, Taiwan; 131624@cch.org.tw (I.-M.H.); leeyulin@gmail.com (Y.-L.L.); chuneng@cch.org.tw (C.-E.L.); 10Division of Infectious Diseases, Department of Internal Medicine, China Medical University Hospital, Taichung 404, Taiwan; cmchid@yahoo.com.tw; 11Division of Critical Care Medicine, Department of Emergency & Critical Care Medicine, Fu Jen Catholic University Hospital, Fu Jen Catholic University, New Taipei City 243, Taiwan

**Keywords:** nutritional assessment, non-*albicans* candidemia, MUST

## Abstract

Several studies have demonstrated that malnutrition is a negative prognostic factor for clinical outcomes. However, there is limited evidence for the effect of malnutrition on clinical outcomes in patients with candidemia. We investigated the relationship between malnutrition and all-cause 28-day mortality among patients with non-*albicans* candidemia. Between July 2011 and June 2014, all adult patients with non-*albicans* candidemia, including *C. tropicalis*, *C. glabrata*, *C. parapsilosis* and so on, were enrolled. The Malnutrition Universal Screening Tool (MUST) scores were used to determine the patients’ nutritional status before the onset of candidemia. A total of 378 patients were enrolled; 43.4% developed septic shock and 57.1% had a high risk of malnutrition (MUST ≥ 2). The all-cause 28-day mortality rate was 40.7%. The Cox proportional hazards model revealed that *C. tropicalis* (HR, 2.01; 95% CI, 1.24–3.26; *p* = 0.005), Charlson comorbidity index (HR, 1.10; 95% CI, 1.03–1.18; *p* = 0.007), Foley catheter use (HR, 1.68; 95% CI, 1.21–1.35; *p* = 0.002), concomitant bacterial infections (HR, 1.55; 95% CI, 1.11–2.17; *p* = 0.010), low platelet count (HR, 3.81; 95% CI, 2.45–5.91; *p* < 0.001), not receiving antifungals initially (HR, 4.73; 95% CI, 3.07–7.29; *p* < 0.001), and MUST ≥ 2 (HR, 1.54; 95% CI, 1.09–2.17; *p* = 0.014) were independently associated with all-cause 28-day mortality. A simple screening tool for nutritional assessment should be used for patients with non-*albicans* candidemia to detect early clinical deterioration, and a tailored nutritional care plan should be established for malnourished individuals, to improve their clinical outcomes.

## 1. Introduction

*Candida*, one of the most frequently encountered invasive fungal infections in humans, causes a wide spectrum of clinical manifestations, including candidemia, intra-abdominal infections, and osteomyelitis [1]. Candidemia usually leads to longer hospital stay, increased medical costs, substantial complications, and all-cause in-hospital mortality of up to 30–60% [2,3]. *Candida albicans* remains the leading cause of candidemia, but non-*albicans* candidemia has progressively increased worldwide in recent years, with a varying distribution of species between different countries, particularly in the Asia-Pacific regions [4,5,6]. Due to the high mortality rate of patients with candidemia, the presence of septic shock [7,8,9,10,11,12], need for mechanical ventilation [7], severe underlying diseases [7,11], multiple organ failure [8], aging [11], concomitant bacterial infections [12], and inappropriate initiation of antifungal agents [9] were found to be independent risk factors associated with mortality in various studies. Furthermore, different fatality rates have been observed based on different *Candida* species, and it seems that patients with non-*albicans* candidemia have a worse prognosis [13]. 

Universally, malnutrition is a key topic: its prevalence is as high as 37.8–78.1% in critically ill patients [14], and around 20–50% of hospitalized patients are diagnosed with undernutrition upon admission [15]. Several prior studies have validated the association between poor nutritional status and adverse clinical outcomes among patients receiving intensive care [14,15,16,17], particularly those with candidemia [18]. Different nutrition assessment and screening tools have been used in the previous studies, including Subjective Global Assessment [19], Mini Nutritional Assessment [20], Nutrition Risk Screening-2002 (NRS-2002) [21], and Malnutrition Universal Screening Tool (MUST) [22]. MUST, the foundation for the NRS-2002, was established to recognize individuals with malnourishment in all healthcare settings. It had three components: calculation of the body mass index (BMI) as well as recording of the percentage of unintentional weight loss in the past months and the presence of acute sickness with a reduction in nutritional intake for days [22]. Limited research has been done on the influence of malnourishment on the clinical outcomes of patients with non-*albicans* candidemia. This study aimed to compare the clinical characteristics of non-*albicans* candidemic patients with a high risk of malnutrition to those of patients without a high risk and antifungal susceptibility, and to identify the predictors of all-cause 28-day mortality among patients with non-*albicans* candidemia.

## 2. Materials and Methods

### 2.1. Study Design and Settings

We conducted a retrospective multicenter observational study at five tertiary hospitals, including Taipei Veterans General Hospital in northern Taiwan, China Medical University Hospital and Changhua Christian Hospital in central Taiwan, Chi Mei Medical Center, Liouying Branch, and Kaohsiung Medical University Hospital in southern Taiwan, between 1 July 2011 and 30 June 2014. All adult patients aged over 20years with positive blood cultures for non-*albicans Candida* species admitted to any of the five participating hospitals were eligible for enrollment. Only the first episode of non-*albicans* candidemia was analyzed in patients who developed two or more episodes of candidemia during the study period. Individuals with two or more non-*albicans Candida* species isolated from blood cultures were excluded. A standardized case report form was used to collect information on the demographics and clinical characteristics, including the age, sex, source of candidemia, Charlson comorbidity index, risk factors (receiving chemotherapy, total parenteral nutrition (TPN), use of central venous catheters, recent abdominal surgery, steroids), disease severity (septic shock and receiving intensive care), concomitant bacterial infections, laboratory data, BMI, initial antifungal treatment, and 28-day mortality. The study was approved by the medical ethics committees of the five participating hospitals, and all clinical investigations were performed following the guidelines of the Declaration of Helsinki.

### 2.2. Definitions

Candidemia was confirmed by the isolation of non-*albicans Candida* species in at least one set of blood cultures among patients with corresponding clinical symptoms/signs of infection [23]. The origin of the candidemia was determined based on clinical manifestations, microbiological findings, and radiological investigations. Catheter-related bloodstream infection (CRBSI) was defined by the same organism being isolated from the indwelling catheter segment with the growth of ≥15 colony-forming units via semi-quantitative tip culture and peripheral blood culture [24]. If no apparent focus of the infection was identified at other sites, candidemia was classified as primary [23]. Septic shock was diagnosed when a patient had either systolic blood pressure ≤ 90 mmHg or mean arterial pressure ≤ 70 mmHg accompanied by the use of vasopressors. A concomitant bacterial infection was defined as a bacterial infection occurring within seven days prior to the first episode of the candidemia. Use of steroids was defined as a prescription of at least 10 mg prednisolone or an equivalent daily dosage for more than one week within 30 days before candidemia [25]. The initial antifungal treatment was the first prescribed antifungal agent, while invasive fungemia was clinically suspected. All-cause 28-day mortality was recorded when a patient died from any cause during hospitalization.

### 2.3. Mycological Diagnosis of Candidemia and Antifungal Susceptibility Testing

Fungal blood cultures were processed using the BACTEC culture system (Becton Dickinson Microbiology System, Sparks, MD, USA). All non-*albicans Candida* isolates were identified to the species level by morphology analysis on CHROMagar (Creative Life Science, Ltd., New Taipei City, Taiwan) and biochemical methods using the API ID 32C system (bioMérieux, Marcy l’Etoile, France) or Vitek 2 system (bioMérieux, Marcy l’Etoile, France) according to the regulations of each participating hospital. The in vitro susceptibility to nine antifungal agents, including anidulafungin, caspofungin, micafungin, fluconazole, voriconazole, itraconazole, posaconazole, 5-fluocytosine, and amphotericin B, was determined using the broth microdilution method with the Sensititre YeastOne system (Trek Diagnostic Systems, Ltd., East Grinstead, UK) according to the manufacturer’s instructions. We utilized *C. krusei* (ATCC 6258) and *C. parapsilosis* (ATCC 22019) as reference isolates for quality control. The minimal inhibitory concentrations (MICs) of the nine tested antifungal agents were interpreted using the clinical breakpoints or epidemiological cut-off values proposed by the Clinical and Laboratory Standard Institute (CLSI) recommendations [26].

### 2.4. Nutrition Assessment

All patients with non-*albicans* candidemia underwent nutritional assessment via measurement of MUST scores before the onset of candidemia. The information needed for the MUST was provided by either the patients or their caregivers. As described before, the MUST methodology comprised three independent parameters: BMI score (BMI > 20 = 0, BMI 18.5–20 = 1, BMI < 18.5 = 2); percentage of unexpected weight loss in the past 3–6 months (weight loss < 5% = 0, weight loss 5–10% = 1, weight loss > 10% = 2); acute disease effect score (a score of 2 was added if the patient was acutely ill with subsequent lack of any dietary intake for more than five days). Based on the total scores calculated, the development of malnourishment was determined as low (total score = 0), medium (total score = 1), and high risk (total scores ≥ 2). We further classified MUST into a dichotomized group (low to medium risk of malnutrition 0 or 1 (non-high-risk group), versus high risk of malnutrition ≥ 2 (high-risk group)). The MUST score was selected as the evaluation tool for this study because of its good predictive value for clinical outcomes in a variety of patient populations [27,28,29]. 

### 2.5. Statistical Analyses 

Descriptive data for quantitative variables were expressed as mean ± standard deviation and analyzed using Student’s *t*-test or Mann–Whitney U test. Categorical variables were presented as counts (%) and compared using the chi-squared test or Fisher’s exact test. Statistical significance was determined using two-tailed tests, and a *p*-value < 0.05 was considered as the threshold for statistical significance. To identify the independent predictors of all-cause 28-day mortality, a Cox proportional hazards model in a stepwise approach was used to calculate the hazard ratios and corresponding 95% confidence intervals. Survival was plotted using Kaplan–Meier curves, and differences were evaluated using the log-rank test. All statistical analyses were conducted using SAS statistical software (version 9.4; SAS Institute, Cary, NC, USA).

## 3. Results

Initially, 597 patients with a diagnosis of non-*albicans* candidemia were recruited, but 219 patients were excluded because of missing clinical data, including 205 with incomplete MUST scores, 12 with incorrect sampling, and two that were transferred to another hospital. Finally, 378 patients were enrolled in this study, and 146 (38.6%) were infected with *C. tropicalis*, 133 (35.2%) with *C. glabrata*, 78 (20.6%) with *C. parapsilosis*, 21 (5.6%) with other *Candida* species, including *C. guilliermondii* (8, 2.1%), *C. krusei* (4, 1.1%), *C. lusitaniae* (2, 0.5%), *C. lipolytica* (2, 0.5%), *C. intermedia* (1, 0.3%), *C. catenulate* (1, 0.3%), *C. metapsilosis* (1, 0.3%), *C. pulcherrima* (1, 0.3%), and *C. sake* (1, 0.3%) (Figure 1).

Among these patients, more than 60% were men, and their mean ± standard deviation of age was 66 ± 17 years (Table 1). The three leading origins of candidemia were the central venous catheter, primary infection, and an intra-abdominal focus. The predominant non-*albicans Candida* species was *C. tropicalis* (38.6%). More than two-fifths (43.4%) of the patients developed septic shock due to candidemia, and the mean ± standard deviation of the Charlson comorbidity score was 4.2 ± 2.5. Previous use of steroids was the most common precipitating factor associated with non-*albicans* candidemia, and approximately half (47.4%) of all the patients were found to have concomitant bacterial infections. The majority (71.2%) of the participants were prescribed fluconazole initially, and the all-cause 28-day mortality rate in this study was 40.7%.

The in vitro susceptibilities of the 304 non-*albicans Candida* isolates to the nine tested antifungal agents are summarized in Table 2. All three echinocandins exhibited excellent sensitivity against these strains (>95%), except for caspofungin against *C. glabrata* isolates (91.8%), indicating that echinocandins remain the drug of choice for treating non-*albicans* candidemia. However, the echinocandin MIC_50_ and MIC_90_ of *C. parapsilosis* (0.5–1 and 0.5–2 mg/L, respectively) were higher than those of the other non-*albicans Candida* species. The overall fluconazole-resistance rate among all non-*albicans Candida* isolates was 10.2% (31), including 25 in *C. tropicalis*, four in *C. glabrata*, and two in *C. parapsilosis*. Moreover, *C. glabrata* displayed higher fluconazole MIC_50_ and MIC_90_ (16 mg/L and 32 mg/L, respectively), and the highest resistance rate to fluconazole was observed in *C. tropicalis* (17.1%). The voriconazole-resistance rates in *C. tropicalis* and *C. parapsilosis* were 15.1% and 2.1%, respectively.

The comparisons of the demographics, laboratory data, and clinical outcomes between the high-risk and non-high-risk groups are also illustrated in Table 1. Half of the patients (57.1%) belonged to the group with a high risk of malnutrition, and the mean age and sex distribution were not different between the two groups. More primary candidemia and catheter-associated sources were found in the non-high-risk group, but the focus of candidemia being the urinary tract, respiratory tract, and intra-abdominal region was more frequently observed in the high-risk group. The patients who acquired infections caused by the three predominant types of the non-*albicans Candida* species were significantly different between the two groups. A higher proportion of patients in the high-risk group presented with septic shock (65.4% vs. 35.8%, *p* = 0.01), but the Charlson comorbidity index score was similar between the two groups. As for the risk factors related to candidemia, only TPN use was much more commonly encountered in high-risk patients, with significant differences. Furthermore, patients at high risk for malnutrition tended to have concomitant bacterial infections. Laboratory parameters did not differ between the two groups, except for a higher platelet count observed in the high-risk group. ICU admission and all-cause 28-day mortality were not different between the two groups, although a slightly higher percentage of patients at high risk for malnutrition died than that of the non-high-risk patients, without a significant difference (59.9% vs. 35.2%, *p* = 0.06). We further explored the 28-day survival probability of high-risk versus non-high-risk patients acquiring non-*albicans* candidemia, and there was a trend that the former had a higher mortality rate when compared to the latter, as shown in Figure 2 (*p* = 0.08).

To verify the importance of nutrition assessment on the clinical outcomes of patients with non-*albicans* candidemia, a Cox proportional hazards model in a stepwise approach was performed using specific identified factors (Table 3). The source of the candidemia, *C. tropicalis*, *Candida* colonization, Foley catheter use, concomitant bacterial infections, lower haemoglobin level, thrombocytopenia, ICU admission, septic shock, and no antifungal therapy initially were all associated with all-cause 28-day mortality in patients with non-*albicans* candidemia. After adjustment for confounders, *C. tropicalis* (HR, 2.01; 95% CI, 1.24–3.26; *p* = 0.005), Charlson comorbidity index (HR, 1.10; 95% CI, 1.03–1.18; *p* = 0.007), Foley catheter use (HR, 1.68; 95% CI, 1.21–1.35; *p* = 0.002), concomitant bacterial infections (HR, 1.55; 95% CI, 1.11–2.17; *p* = 0.010), low platelet count (HR, 3.81; 95% CI, 2.45–5.91; *p* < 0.001), not receiving antifungals initially (HR, 4.73; 95% CI, 3.07–7.29; *p* < 0.001), and MUST ≥ 2 (HR, 1.54; 95% CI, 1.09–2.17; *p* = 0.014) were recognized as independent predictors of a higher risk of all-cause 28-day mortality.

## 4. Discussion

An increasing number of studies have demonstrated that non-*albicans Candida* has played an increasingly clinical role in the recent decades [4,5,6,30,31]. Additionally, malnutrition might predict a poorer prognosis in critically ill patients, including patients with candidemia [18]. To the best of our knowledge, this is the first study to authenticate the risk factors for all-cause 28-day mortality among non-*albicans* candidemic individuals and the impact of undernutrition on clinical outcomes. In the present study, the majority of the non-*albicans Candida* species were *C. tropicalis*, and 43.4% of the patients developed septic shock with an all-cause 28-day mortality rate of 40.7%. Echinocandins retained their excellent in vitro susceptibility against non-*albicans Candida* isolates, and the overall fluconazole-resistance rate was approximately 10%. More than 50% of the candidemic patients were classified as having a high risk of malnutrition, and the independent predictors of all-cause 28-day mortality were *C. tropicalis*, Charlson comorbidity index, Foley catheter use, concomitant bacterial infections, low platelet count, no antifungal use initially and MUST ≥ 2. 

Most recent studies have mainly analyzed the predictors of mortality in candidemic patients with secondary results regarding the incidence of septic shock, which ranged from 21.2 to 49.3% in candidemic individuals due to all *Candida* species [10,12,32,33], and 28.6–31.6% among patients with non-*albicans Candida* bloodstream infections [33,34,35]. However, a much higher prevalence of septic shock (43.4%) was observed in our study, probably because more patients with *C. tropicalis* candidemia were enrolled, which was supported by Ko et al.’s proposal [36]. Additionally, the overall fluconazole-resistance rate and in vitro susceptibility rate of echinocandins against non-*albicans Candida* isolates were, 3.4–8.8% [1,12,37] and 97.6–100% [1,37], respectively, comparable to our findings. Regarding the clinical outcomes, a few recent reports illustrated in-hospital mortality rates of approximately 44.6–60.0% [7,32,38], similar to our results, implicating a poor prognosis in patients with non-*albicans Candida* bloodstream infections. 

As previously reported, patients with fumgemia due to *C. tropicalis* might have a higher mortality rate than those with fungemia due to other *Candida* species [39,40]. Additionally, the significant association of *C. tropicalis* with poor clinical outcomes has been validated [36], which is consistent with our findings. Possible explanations for the virulence mechanisms of *C. tropicalis,* including its adhesion behavior, biofilm formation, hemolysis activity, secretion of enzymes, and dissemination potential [41], have been elucidated, and prominent resistance to azoles by *C. tropicalis* affecting the initiation of the appropriate antifungals might be another reason from our study. Therefore, clinicians should be aware of the care of patients with *C. tropicalis* fungemia.

Patient-related factors, such as comorbid conditions and a poor prognosis, are highly relevant among individuals with *Candida* bloodstream infections. Notably, the Charlson comorbidity index, a comorbidity measure useful for predicting mortality risk, has been demonstrated in several studies [11,42], comparable to our results. Likewise, indwelling urinary catheters as a risk factor for mortality among candidemic patients in our study has also been previously illustrated [43,44]: predisposition to urinary tract infections and formation of biofilms on the surface of catheters lowering the efficacy of antifungals resulted in patients being more susceptible to death. Subsequently, detailed calculation of the Charlson comorbidity index and early removal of unnecessary urinary catheters would aid healthcare workers in handling the clinical outcomes of non-*albicans* candidemia cases.

The role of concurrent bacterial infections on in-hospital mortality in patients with candidemia has been established in previous reports [12,45], similar to our results. It was anticipated that concomitant bacterial infections would lead to candidemic patients being vulnerable to severe sepsis or septic shock and eventually increase the risk of mortality; therefore, effective antimicrobial agents against concurrent bacterial infections should be initiated aggressively to improve the clinical prognosis. Moreover, thrombocytopenia has been proposed as implicating candidemic patients with more seriously ill conditions and at high risk of mortality [46]. The adverse effects of low platelets on patients’ outcomes in response to fungal infection [12] and the key role of platelets in inflammation, immunity, and infectious diseases [47] might be the causes. Nevertheless, the molecular mechanisms of thrombocytopenia and platelet function in non-*albicans* candidemia should be further studied. Undoubtedly, early administration and appropriate initiation of antifungal agents for fungemic patients will lead to a better prognosis [9,48] than if no antifungals are initiated for these patients from the start, similar to our findings. Consequently, empirical or preemptive antifungal therapy should be considered prudently when clinicians are highly suspicious of non-*albicans* candidemia.

To the best of our knowledge, no previous research has explored the relationship between malnutrition and clinical outcomes in patients with non-*albicans* candidemia. Nutritional deficiencies are linked to a higher risk of infections and mortality from bloodstream infections [49]. A few studies have demonstrated that malnutrition predisposes acutely ill patients to acquiring infections [14], probably due to the persistent inflammation and immune dysfunction associated with undernutrition [50], and eventually, there can be progression to severe sepsis, septic shock, and death. Hence, poor nutritional status as an independent factor for all-cause 28-day mortality in patients with non-*albicans* candidemia is reasonable. Malnutrition is a modifiable factor: its early identification and correction among patients at high-risk might prevent its potential deleterious effects, even though there are no internationally standardized screening tools for the assessment of nutritional status [51]. MUST has been validated to predict the risk of malnutrition with the highest sensitivity and hazard ratio of mortality by Rabito et al. [27], which was similar to our results. Furthermore, non-*albicans* fungemic individuals within the high-risk group tended to have a higher mortality rate via the Kaplan–Meier method. Therefore, routine nutrition screening for all hospitalized patients, particularly those with non-*albicans* candidemia, not only predicted the risk of all-cause 28-day mortality but also established an adequate nutritional care plan tailored to these malnourished individuals to lower complication rates.

There were some limitations to our study. First, it was conducted retrospectively; therefore, unavoidable bias and missing data are anticipated. Second, there could have been trivial variations in the collection of MUST data because this was a multicenter study, and measurement of weight loss percentage was probably a barrier during clinical practice. Finally, serum levels of some inflammatory mediators, such as interleukin-6 or interleukin-10, were not tested in our study, and how inflammatory cytokines in malnutrition patients with non-*albicans* candidemia played a role in the risk of mortality requires further investigation.

## 5. Conclusions

Our study revealed that a high risk of malnourishment, that is, MUST ≥ 2, is independently associated with a higher risk of all-cause 28-day mortality in non-*albicans* candidemic patients. A simple and rapid screening tool for malnutrition risk, such as MUST, can help physicians recognize malnourished individuals, and early goal-directed therapy for high-risk patients can be achieved as soon as possible. Furthermore, because malnutrition remains a reversible and treatable condition, an optimal nutritional care strategy should be implemented appropriately to improve patients’ prognosis. 

## Figures and Tables

**Figure 1 nutrients-13-03218-f001:**
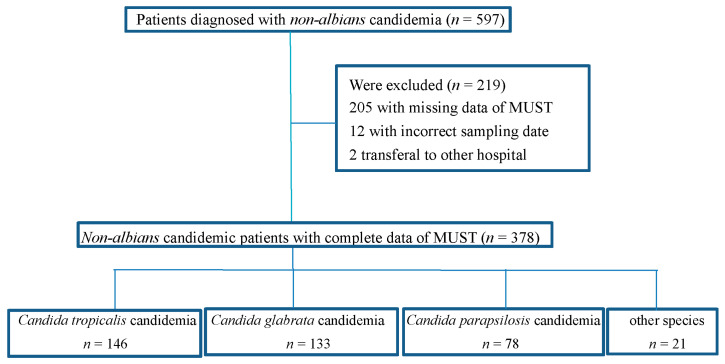
Flow chart of patients with *non-albicans* candidemia included in the analysis.

**Figure 2 nutrients-13-03218-f002:**
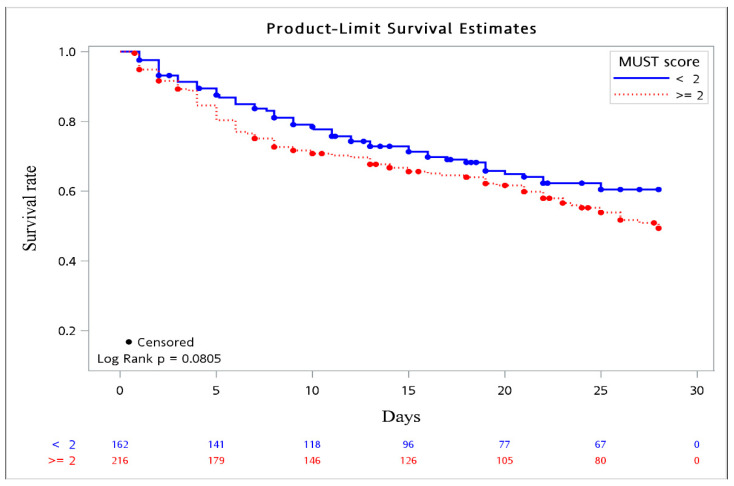
Kaplan–Meier survival curve. Survival probability based on MUST score (≥2 vs. <2).

**Table 1 nutrients-13-03218-t001:** Comparison of demographics and clinical characteristics of candidemic patients with MUST ≥ 2 and <2.

	All, %	MUST ≥ 2	MUST < 2	*p*
	(*n* = 378)	(*n* = 216)	(*n* = 162)	
Age	66.13 ± 16.88	65.76 ± 16.79	66.63 ± 17.04	0.62
Gender (male)	235 (62.2)	137 (63.4)	98 (60.5)	0.560
Source of candidemia				<0.001
Primary	89 (23.5)	33 (15.3)	56 (34.6)	
Central catheter	100 (26.5)	57 (26.4)	43 (26.5)	
Urinary tract	56 (14.8)	38 (17.6)	18 (11.1)	
Respiratory tract	61 (16.1)	39 (18.1)	22 (16.2)	
Intra-abdomen	65 (17.2)	47 (21.8)	18 (11.1)	
Wound	4 (1.1)	1 (0.5)	3 (1.9)	
*Candida* species				0.01
*C. tropicalis*	146 (38.6)	89 (41.2)	57 (35.2)	
*C. glabrata*	133 (35.2)	82 (38.0)	51 (31.5)	
*C. parapsilosis*	78 (20.6)	33 (15.3)	45 (27.8)	
Others	21 (5.6)			
Charlson comorbidity index	4.23 ± 2.48	4.19 ± 2.47	4.29 ± 2.49	0.73
Candida colonization	133 (35.2)	84 (38.9)	49 (30.2)	0.08
Receiving chemotherapy	106 (28.0)	59 (27.3)	47 (29.0)	0.72
TPN use	82 (21.7)	57 (26.4)	25 (15.4)	0.01
Use of venous access	267 (70.6)	160 (74.1)	107 (66.0)	0.09
Foley use	170 (45.0)	100 (46.3)	70 (43.2)	0.55
Recent abdominal surgery	63 (16.7)	43 (19.9)	20 (12.3)	0.05
Use of steroids	309 (81.7)	174 (80.6)	135 (83.3)	0.49
Concomitant bacterial infection	179 (47.4)	114 (52.8)	65 (40.1)	0.01
WBC (/µL)				0.35
<6700	125 (33.1)	65 (30.1)	60 (37.0)	
≥6700 and < 12,100	124 (32.8)	73 (33.8)	51 (31.5)	
≥12,100	129 (34.1)	78 (36.1)	51 (31.5)	
Hb < 10 mg/dl	209 (55.3)	115 (53.2)	94 (58.0)	0.35
Platelet (/µL)				0.15
<79,000	133 (35.2)	67 (31.0)	66 (40.7)	
≥79,000 and <185,000	120 (31.7)	73 (33.8)	47 (29.0)	
≥185,000	125 (33.1)	76 (35.2)	49 (30.2)	
ICU admission	61 (15.1)	39 (18.1)	22 (13.6)	0.24
BMI	21.96 ± 4.57	20.45 ± 4.41	23.98 ± 3.98	<0.001
Septic shock	164 (43.4)	106 (65.4)	58 (35.8)	0.01
Antifungal therapy				0.19
Initial use of fluconazole	269 (71.2)	157 (72.7)	112 (69.1)	
Initial use of echinocandin	65 (17.2)	31 (14.4)	34 (21.0)	
None	44 (11.6)	28 (12.3)	16 (9.9)	
Death within 28 days	154 (40.7)	97 (59.9)	57 (35.2)	0.06

**Table 2 nutrients-13-03218-t002:** In vitro susceptibility of antifungal agents among 304 non-*albicans Candida* isolates.

Species	Antifungal Agent	MIC (mg/L)			Susceptibility, No. (%)		
		Range	MIC_50_	MIC_90_	S	SDD	R (I + R)
*C. tropicalis*	Anidulafungin	0.03–1	0.12	0.25	142 (97.3)	-	3 (2.1)
(*n* = 146)	Caspofungin	0.015–>8	0.12	0.25	141 (96.6)	-	5 (3.4)
	Micafungin	0.015–2	0.03	0.03	142 (97.3)	-	4 (2.7)
	Fluconazole	0.25–>256	2	16	94 (64.4)	27 (18.5)	25 (17.1)
	Voriconazole	0.015–>8	0.25	2	65 (44.5)	59 (40.4)	22 (15.1)
	Itraconazole	0.06–1	0.25	0.5	-	-	-
	Posaconazole	0.015–1	0.25	0.5	-	-	-
	Flucytosine	≤0.06–>64	≤0.06	0.12	-	-	-
	Amphotericin B	≤0.12–2	0.5	1	-	-	-
*C. glabrata*	Anidulafungin	0.03–1	0.06	0.12	109 (99.1)	-	1 (0.9)
(*n* = 110)	Caspofungin	0.03–1	0.12	0.12	101 (91.8)		9 (8.2)
	Micafungin	≤0.008–1	0.015	0.015	108 (98.2)		2 (1.8)
	Fluconazole	0.5–>256	16	32	-	106 (96.4)	4 (3.6)
	Voriconazole	≤0.008–4	0.5	0.5	-	-	-
	Itraconazole	≤0.015–4	0.5	1	-	-	-
	Posaconazole	≤0.008–>8	1	2	-	-	-
	Flucytosine	≤0.06–0.5	≤0.06	≤0.06	-	-	-
	Amphotericin B	≤0.12–4	0.5	1	-	-	-
*C. parapsilosis*	Anidulafungin	0.25–2	1	2	48 (100)	0 (0)	0 (0)
(*n* = 48)	Caspofungin	0.25–1	0.5	0.5	48 (100)	0 (0)	0 (0)
	Micafungin	0.25–2	1	2	48 (100)	0 (0)	0 (0)
	Fluconazole	0.25–16	1	2	45 (93.8)	1 (2.1)	2 (4.2)
	Voriconazole	≤0.008–0.5	0.015	0.06	46 (95.8)	1 (2.1)	1 (2.1)
	Itraconazole	0.03–1	0.12	0.12	-	-	-
	Posaconazole	0.015–1	0.03	0.06	-	-	-
	Flucytosine	≤0.06–0.5	0.12	0.25	-	-	-
	Amphotericin B	≤0.12–1	0.5	0.5	-	-	-

**Table 3 nutrients-13-03218-t003:** Predictors associated with 28-day mortality among patients with non-*albicans* candidemia.

Variables	Unadjusted HR	*p*-Value	Adjusted HR	*p*-Value
	(95% CI)		(95% CI)	
Age	1.00 (0.99–1.01)	0.943		
Gender, male	1.24 (0.89–1.73)	0.208		
Source of candidemia				
Primary		Reference		
Central catheter	2.35 (1.42–3.89)	0.001		
Urinary tract	1.10 (0.15–8.16)	0.925		
Respiratory tract	1.96 (1.10–3.50)	0.022		
Intra-abdomen	2.19 (1.26–3.81)	0.005		
Wound	2.43 (1.43–4.12)	0.001		
*Candida* species				
*C. parapsilosis*		Reference		Reference
*C. tropicalis*	2.36 (1.47–3.79)	<0.001	1.83 (1.12–2.97)	0.015
*C. glabrata*	1.30 (0.79–2.16)	0.305	0.92 (0.55–1.56)	0.766
Others	1.20 (0.51–2.80)	0.681	1.09 (0.46–2.60)	0.839
Charlson comorbidity index	1.07 (1.00–1.14)	0.052	1.08 (1.01–1.16)	0.020
Candida colonization	1.38 (1.00–1.90)	0.048		
Receiving chemotherapy	0.79 (0.55–1.15)	0.222		
TPN use	0.93 (0.64–1.35)	0.687		
Use of venous access	1.12 (0.79–1.60)	0.535		
Foley use	1.65 (1.20–2.26)	0.002	1.67 (1.19–2.35)	0.003
Recent abdominal surgery	0.94 (0.62–1.43)	0.775		
Use of steroids	1.39 (0.88–2.20)	0.161		
Concomitant bacterial infection	1.60 (1.16–2.20)	0.004	1.49 (1.06–2.08)	0.021
WBC (/µL)				
<6700		Reference		
≥6700 and <12,100	0.74 (0.50–1.11)	0.143		
≥12,100	0.93 (0.64–1.36)	0.722		
Hb <10 mg/dL	1.54 (1.11–2.15)	0.010		
Platelet (/µL)				
<79,000	3.70 (2.41–5.69)	<0.001	3.98 (2.56–6.19)	<0.001
≥79,000 and <185,000	1.72 (1.07–2.76)	0.025	1.58 (0.97–2.55)	0.064
≥185,000		Reference		Reference
ICU admission	1.48 (1.01–2.18)	0.045		
BMI	1.00(0.96–1.03)	0.780		
Septic shock	2.06 (1.49–2.84)	<0.001		
Antifungal therapy				
Initial use of fluconazole		Reference		Reference
Initial use of echinocandin	1.02 (0.65–1.59)	0.927	1.02 (0.65–1.61)	0.932
None	4.94 (3.25–7.51)	<0.001	4.73 (3.07–7.29)	<0.001
MUST ≥ 2	1.33 (0.96–1.85)	0.085	1.46 (1.03–2.07)	0.034

## Data Availability

The datasets used during the current study are available from the corresponding author on reasonable request.

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
