# Peer review of "Impact of Nutritional Assessment on the Clinical Outcomes of Patients with Non-albicans Candidemia: A Multicenter Study"

_nutrients, 2021, doi:10.3390/nu13093218_

Round 1
Reviewer 1 Report
Dear Author,
Thank you for submitting manuscript titled “Impact of nutritional assessment on the clinical outcomes of patients with non-albicans candidemia: a multicenter study”
First of all, I congratulate for your success of great work. I have carefully read and evaluated this manuscript of clinical experience, and I have only minor comments for acceptance of Nutrients.
Thank you for your consideration. If you have any questions and problems, please feel free to ask me ASAP.
Reviewer comments
This manuscript is well written about the clinical outcomes of patients with non-albicans candidemia. I believe that this novel research will be of great importance for the clinical outcomes of patients with fungal infection. Please revise this manuscript following these comments as below.
Minor comments:
Manuscript will be fine, I have only comment that “Introduction” might be too long and redundant. Please make shorter and easy to read.
Reviewer 2 Report
Yi-Chien Lee et al have reported very interesting data assessing the association between nutritional status and the clinical outcomes of patients with non-albicans candidemia
The introduction is well detailed with background information on candidemia and its epidemiology burden
1-Line 72. Please provide the definition of MUST abbreviation
Method
This session is well structured and the statical analysis detailed
2-The study design and procedure were well described, however, I would suggest to the authors to draft the flowchart figure to facilitate understanding of the participant selection and inclusion procedure.
-Result
3-Lines 163-166 please add the percentage
4-I did not see the tables and figures please send it
5-I did not see the Kaplan Meir survival analysis curve interpretation,please check
